# Utilization of Fluidized Bed Combustion Fly Ash in the Design of Reuse Clay Soil in the Form of Self-Compacting Grouts

**DOI:** 10.3390/ma13081972

**Published:** 2020-04-23

**Authors:** Rostislav Drochytka, Magdaléna Michalčíková

**Affiliations:** Faculty of Civil Engineering, Brno University of Technology, Veveří 331/95, 602 00 Brno, Czech Republic; drochytka.r@fce.vutbr.cz

**Keywords:** clay soil, recycling, cement, self-compacting grout, fluidized bed combustion fly ash (FBCA), sodium carbonate

## Abstract

This paper addresses the influence of fluidized bed combustion fly ash (FBCA) and further liquefying additives on the formation of structure and on the resulting properties of self-compacting grouts based on clay soil. In order to give the best account of the influence of individual input materials, tests were conducted on samples without the use of fluidized bed combustion fly ash. Clay soil (Cl) and cement were used as input materials, and fluidized bed combustion fly ash (10% and 30%) and a liquefying additive (sodium carbonate 0.1%) were used as an admixture. It has been experimentally determined that the use of 10% FBCA with clay soil is most suitable for achieving the optimal spillage parameter of self-compacting grout (class SF2 (660–750 mm) and class SF3 (760–850 mm)). It was also found that fluidized bed combustion fly ash and the liquefying additive have a significant influence on the formation of the structure of the self-compacting grout and, due to their presence, the compressive strength of the samples increased up to 0.5 MPa after seven days of hardening. The reaction between 0.1% of sodium carbonate and clay soil increased the electrokinetic potential, which reduced the viscosity of the self-compacting grout. Within the research work, the verification of the developed self-compacting grout in situ was also carried out.

## 1. Introduction

This paper researches the usability of clay soils that arise in the framework of excavation works of utilities (underground utilities UU). Attention to this material was given in the paper mainly because the production of unsuitable (mainly clay) soils from UU excavation works occupies a significant part. From of all types of waste, it is about 65% [1,2,3]. In view of the incoming new legislative requirements, and especially under the Waste Framework Directive [4], the aim was to find a solution to the problem, not only by minimizing the formation of these soils, but also in their use for subsequent building production [5]. In fact, this directive requires member states of the European Union to achieve a recycling target of 70% by 2020 [4].

Under current legislation [4,5], soils produced under the UU are treated as waste. This is due to the properties of clay soils and because they often become useless at the construction site. Among the undesirable properties of soils are, in particular, their sensitivity to liquefaction, softness, content of higher amounts of organic substances, etc. [6,7,8]. For these reasons, it is not possible to reuse them in the construction and, therefore, they often end up in landfills, waste dumps, or dump areas.

Generally speaking, from a geotechnical point of view, it is impossible to obtain soil that meets the requirements for its re-use in the UU excavation without any modification. For this reason, the aim is to find optimal technologies or methods that lead to the improvement of the required soil parameters so that it can be reused in the construction. When choosing the technology, it is always necessary to discern what purpose the soil will serve. In today’s practice there are many technologies available for soil modification or treatment [9,10,11]. In this paper, the method of improving the properties of clay soils focused on is chemical stabilization by FBCA and their subsequent liquefaction.

Chemical stabilization aims to modify the soil moisture and, above all, to improve its strength, which increases the soil’s softening resistance [12,13,14,15,16,17]. The amount and type of FBCA added should in fact be consistent with the clay mineral content of the soil [11,18,19,20,21,22]. Exact values of the amount and determination of the effect of FBCA on clay soil can be determined after the implementation of laboratory tests.

Non-traditional possibilities of clay soil treatment include liquefaction into a self-compacting grout [23,24,25]. This is a new type of technology that is not widely known in the current practice. The principles of liquefaction of clay soils are based on the use of FBCA and liquefying additives that are willing to react with clay minerals, which mainly results in improved spillage and a reduced amount of mixing water in the design of self-compacting grouts [24]. Since this is a relatively unknown topic, the use of a liquefying additive was based on experience in the manufacture of ceramic products.

From the above findings, the aim of the paper is the research and development of clay soil reuse, along with FBCA and further liquefying additives, in the form of self-compacting grouts for UU excavations, road substructures, or other building constructions, which will achieve high competitiveness on domestic and foreign markets.

## 2. Materials and Methods

The main raw material for the design of self-compacting grouts was soil, which arises in the framework of UU excavation works. Due to the nature of soils and location, the Central European region (Brno, Czech Republic-CZ) was chosen for experimental activities. It is a typical locality in which there are almost all types of soil from limestones, clays, loesses, to sands [26]. The specific clay soil, marked ‘Cl’ according to classification in accordance with EN ISO 14688-1 [27], was taken from the Carpathian system, where Jurassic and Cretaceous sediments occur. Geologically, these are the so-called flysch parts, which are composed of sandstones to conglomerates with clay slates to claystones, i.e., layers with variable permeability and different strength character (Figure 1). The main parameter of the soil that is monitored with respect to the possibility of its reuse in excavation works is granulometry and mineralogical composition. The mineralogical composition was measured by X-ray diffraction analysis. The measurements of X-ray diffraction analysis were performed at diffraction angle 2θ within a range from 5° to 50° and with Cu Kα radiation. The quantity of the main mineralogical phases of the samples was determined with the Rietveld method and according to the internal standard CaF_2_. Other parameters determined for the soil were liquidity limit w_L_ (54.00%), plasticity limit w_p_ (29.90%), plasticity index I_p_ (24.10%), soil moisture w (25.40%), and degree of consistency I_c_ (1.19). Other parameters of the clay soil were measured according the standards EN ISO 14688-1 [27], EN ISO 15688 [28], EN 1997-2 [29], EN ISO/CD 20500-3 [30], and in particular EN ISO 15176 [31,32]. The results are described in Section 3.

For the mechanical treatment of this clay soil Cl it would be possible to use an admixture of other soils or materials (e.g., fly ashes) of suitable grain size and moisture. In this way, in particular, an improvement in moisture and an adjustment of the granulation curve would be achieved [33]. In fact, the percentage of the individual components in the soil has a fundamental influence on its mechanical properties [34]. For this reason, FBCA, whose chemical composition is shown in Table 1, was used as an additional raw material. With regard to monitoring the influence of fly ash on soil properties adjustment, resulting self-compacting grout, and environmental aspect, the use of FBCA in amounts of 10% and 30% were chosen as optimal (based on soil weight).

With regard to the composition, moisture content, fine particle content, and based on standard requirements, cement stabilization has been selected as a further clay soil Cl treatment process. Specifically, Portland mixed cement CEM II/B-M (S-LL) 32.5 R (hereafter also referred to as CEM) was used. Lime was not used to design the mixture because some lime content in the fluidized fly ash (FBCA) was assumed. With regard to the results of laboratory testing of soils, especially results of plasticity index I_p_ and content of fine-grained fraction f of soils, the use of 4% cement CEM II/B-M (S-LL) 32.5 R (by weight of soil) was chosen as optimal for the design of the formulas. This type of soil treatment has been chosen because, with a suitably selected type and amount of stabilizing agents (lime, cement), improved workability, compactability, frost susceptibility, moisture, compressive strength, and other soil properties can be achieved [35,36]. At the same time, stabilizing additives are materials that are willing to react with soils and can, therefore, be used in the design of self-compacting grout formulas [37,38,39,40].

The last step of the treatment of clay soil Cl into a self-compacting grout was its liquefaction. This treatment step is especially important to achieve easy flow of the grout around the UU in the trench, without the need for vibration. In this case it is a completely new technology of soil treatment, which is not yet commonly used in current practice and is not very well known. Due to the clay mineral content of clay soil Cl, which is similar in composition to clays of ceramic slips, and based on laboratory testing of various types and quantities of liquefying additives. For the particular soil type (clay soil–‘Cl’), the following liquefying additives were selected for verification: sodium hexametaphosphate, sodium tripolyphosphate and sodium carbonate. These liquefying additives were verified in amounts of 0.04%, 0.06%, 0.08%, 0.1%, 0.5%, 1.5%, 2.5%, 3.5%, 5.0%, and 8.0% by weight of soil (see in Figure 2). When comparing the effect of plasticizers on soil liquefaction, it can be seen in Figure 2 that the sodium carbonate had the greatest effect on liquefaction (spillage value). This effect was apparent even if the small amount (0.04% of the soil) of the additive was added. The required spillage (200–300 mm) was achieved with a water content of 48.5%. By using this additive, it has been possible to reduce the amount of water for optimal spillage by 5%. This is why 0.1% of sodium carbonate for the design of self-compacting grouts was chosen as optimum.

For the designed self-compacting grouts, the research work primarily evaluated the physical-mechanical properties as the most important parameters, verifying both the behavior of the mixture in a fresh and a hardened state [41]. The verification of the properties of self-compacting grouts was based on the test methods of EN 12350-8 Testing fresh concrete—Part 8: Self-compacting concrete—slump-flow test [42], and EN 12390-3 Testing hardened concrete—Part 3: Compressive strength of test specimens [43].

The slump-flow test (SF) was performed on freshly mixed self-compacting grout, collected after five minutes of intensive mixing of the components in the mixer. The self-compacting grout was poured into the Abrams cone using a trowel or other available container. The test vessel was then lifted, and the grout spilled over the area. The diameter of the spilled grout in two mutually perpendicular axes was measured using a meter. The result of the Slump-flow test is the average of d_1_ and d_2_ measurements (see Equation (1)), rounded to the nearest 10 mm:(1)$$\mathrm{SF}=\frac{\left(d_{1}+d_{2}\right)}{2}\;\left(\mathrm{mm}\right)$$
where SF: Slump-flow (mm), d_1_: maximum spill diameter (mm), d_2_: spill diameter in a direction perpendicular to d_1_ (mm).

According to the EN 12350-8 [42] the class values of slump-flow (SF) test divided to SF1 (550–650 mm), SF2 (660–750 mm), and SF3 (760–850 mm).

The aim is to achieve optimum spillage when designing self-compacting grouts, i.e., SF2 (660 750 mm) and SF3 (760–850 mm) classes for easier assumption of spillage and flow of UU grout in the trench.

To determine the compressive strength, it was necessary to take the fresh mixture into pre-prepared molds, which were then stored in a laboratory environment where the mixture matured for seven and 28 days. After this time, the compressive strength test was carried out on specimens (hardened cubes of 100 × 100 × 100 mm). The resulting compressive strength of the test specimens was calculated from the values obtained according to the following Equation (2):(2)$$f_{c}=\frac{F}{A_{c}}\;\left(\mathrm{MPa}\right)$$
where f_c_: compressive strength (MPa; N∙mm^−2^), F: maximum failure load (N), A_c_: cross-sectional area of the test specimen on which the compressive load is applied, calculated from the measured dimensions of the specimen (mm^2^).

Based on standard requirements, it is recommended that excavable self-compacting grouts have a compressive strength after three days of maturation greater than 0.14 MPa, and after 28 days of maturation greater than 0.2 MPa. The maximum compressive strength that self-compacting grouts can achieve after 28 days of maturation is recommended at 2.1 MP.

## 3. Results and Discussion

The main raw material was clay soil. For the identification of this material it was necessary to do its testing. In the following text are the results of the parameters of clay soil. The granulometry of clay soil Cl is shown by the granulation curve in Figure 3. From the granulation curve it was possible to read the proportions of the fractions of fine grain f (96.99%) with grains smaller than 0.063 mm, sandy fractions s (2.68%) with grains from 0.063 to 2 mm, and gravel fraction g (0.33%) with grains from 2 mm up to 60 mm. Based on the proportions of individual fractions, it was also possible to specify the soil name to F7 = MH. According to this designation, it was clay with high plasticity class F7. Based on X-ray diffraction analysis (Figure 4) it was found that the following types of minerals mainly occur in the soil: kaolinite, illite, micas, quartz, calcite and montmorillonite. Orthoclase and albite were also recorded in the soil.

Based on geological soil testing, a number of parameters were identified, which made it possible to suggest whether the soil, in the state in which it was excavated, could be reused in the structure, or if further treatment is necessary. Taking into account the standard requirements of EN ISO 14688-1 [27], EN ISO 15688 [28], EN 1997-2 [29], EN ISO/CD 20500-3 [30], and in particular EN ISO 15176 [31,32], the evaluation of clay soil Cl parameters was initially focused on its pre-treatment. The pre-treatment of soils concerns, in particular, those which, according to EN ISO 15688 [28], must not be used in the earthwork. According to the standard, these are organic soils, muds, peat, humus, topsoil, etc. The problem with these soils is primarily that they contain unsuitable fractions, organic substances in quantities greater than 6%, or are extremely plastic. The results of the evaluation of clay soil showed that it does not contain unsuitable fractions, organic substances in the amount of more than 6%, is not contaminated or soaked in any way and therefore its pre-treatment is not necessary. However, based on the evaluation of other parameters of clay soil Cl (liquidity limit w_L_, plasticity limit w_p_, plasticity index I_p_, soil moisture w, and degree of consistency I_c_), according to EN ISO 15688 [28], EN 1997-2 [29], ISO/CD 20500-3 [30], and EN ISO 15176 [31,32] standard requirements, it was found that this soil belongs to the category of ‘unsuitable’ soils (S-F, MH, Cl), which are not intended for direct use without treatment. Given the nature of the soil, existing treatment technologies and its use as self-compacting grout, its mechanical treatment, stabilization, and subsequent liquefaction have been proposed.

### 3.1. Physical–Mechanical Parameters

In the framework of research works of this paper, the main objective was to monitor the effect of FBCA on the resulting properties of self-compacting grouts. The FBCA was used because it was expected to have a positive effect on the adjustment of the properties of self-compacting grouts, both in a fresh and a hardened state. In particular, FBCA was expected to have an effect of free lime on stabilizing the mixture, increasing compressive strength and reducing shrinkage. In addition to the effect of FBCA, a liquefying additive sodium carbonate (SC) was used in the mixtures. This additive was used because, due to its presence, the self-compacting grout should have a lower moisture content, a higher spillage value and a higher compressive strength. Based on the raw materials used, the following formulas have been designed, which are listed in following text and the following results have been achieved (see in Figure 5 and Figure 6). Composition and designation of designed self-compacting grouts:Clay soil (Cl)-Cl-REF;Clay soil (Cl) + 4% CEM II/B-M (S-LL) 32,5 R (CEM)-Cl + 4% CEM;Clay soil (Cl) + 0.1% sodium carbonate (SC)-Cl + 0.1% SC’Clay soil (Cl) + 10% fluidized bed combustion fly ash (FBCA)-Cl + 10% FBCA;Clay soil (Cl) + 10% fluidized bed combustion fly ash (FBCA) + 4% CEM II/B-M (S-LL) 32,5 R (CEM)-Cl + 10% FBCA + 4% CEM;Clay soil (Cl) + 10% fluidized bed combustion fly ash (FBCA) + 4% CEM II/B-M (S-LL) 32,5 R (CEM) + 0.1% sodium carbonate (SC)-Cl + 10% FBCA + 4%CEM + 0.1% SC;Clay soil (Cl) + 30% fluidized bed combustion fly ash (FBCA)-Cl + 30% FBCA;Clay soil (Cl) + 30% fluidized bed combustion fly ash (FBCA) + 4% CEM II/B-M (S-LL) 32,5 R (CEM)-Cl + 30% FBCA + 4% CEM; andClay soil (Cl) + 30% fluidized bed combustion fly ash (FBCA) + 4% CEM II/B-M (S-LL) 32,5 R (CEM) + 0.1% sodium carbonate (SC)-Cl + 30% FBCA + 4%CEM + 0.1% SC.

The following figures (Figure 5 and Figure 6) show the resulting values of the spillage and compressive strength measurements obtained after seven and 28 days of maturation.

From the results of thes-flow test, it can be seen (see in Figure 5) that the addition of FBCA to clay soil Cl significantly influenced the rheological properties of the grout. This fact was already evident when 10% FBCA was added to Cl soil (Cl + 10% FBCA). With this self-compacting grout containing 10% fluid fly ash, a spillage of 710 mm (SF2) at a humidity of 57% was achieved. This phenomenon was influenced by the ability of FBCA grains to facilitate friction over grains of clay minerals, which had an effect on the higher spillage value. Compared to the reference self-compacting grout (Cl-REF), there was more spillage (by 20 mm), but at a higher amount of moisture (by 2%). The higher amount of moisture was affected by the presence of CaO in the fluidized fly ash, which bound water due to the immediate reaction of CaO with water. This reaction is a process of hydration of CaO + H_2_O = Ca(OH)_2_ + 15.5 kcal, evaporation of a certain amount of water and chemical reaction by flocculation of clay particles. Scanning electron microscope (SEM) images show changes in the shape, or microstructure, of the self-compacting grouts (Figure 7a,b)). Chemically it is the so-called grout stabilization that can reduce the amount of moisture if the soil in its natural state contains a high amount of moisture. For a grout containing 30% FBCA (Cl + 30% FBCA), the spillage value decreased to 685 mm (SF2, at 60% humidity), compared to the reference grout (Cl-REF = 690 mm, SF2, at humidity 55%). This phenomenon was influenced by the higher presence of CaO in the fly ash, which immediately reacted with the added water as described above. In general, it can be stated that the addition of FBCA to the mixture, especially in an amount of up to 10%, resulted in an increase of the spillage value and a higher addition (30% of the weight of the soil) stabilized the mixture.

Cement also played a role in evaluating the rheological properties of the self-compacting grout. The cement readily reacted with the clay soil Cl, which was manifested in particular by reducing the sedimentation of the particles while mixing the mixture. In the fresh state, the cement thus proved to be an effective soil stabilizer. From Figure 5, we can see that the presence of cement could reduce the spillage value (Cl + 4% CEM = 700 mm, SF2, at 60% humidity; Cl + 10% FBCA + 4% CEM = 710 mm, SF2, at 57% humidity; Cl + 30% FBCA + 4% CEM = 685 mm, SF2, at 61% humidity), because hydration occurred immediately when water and cement were mixed with FBCA and clay soil Cl. However, in terms of the amount of cement (4% of the weight of the soil) in the case of the designed self-compacting grouts, the spillage value was more influenced by FBCA than by cement.

When evaluating the effect of 0.1% (of the weight of the soil) sodium carbonate liquefying additive on the liquefaction effect of self-compacting grout, it can be stated that the greatest effect was observed with the Cl + 0.1% SC grout, i.e., when no FBCA was used in the grout. The grout reached a spillage value of 705 mm, SF2, at 55% humidity. Compared to the reference grout (Cl-REF = 690 mm, SF2, at 55% humidity), it can be stated that the spillage was improved by 15 mm while maintaining 55% humidity. Thus, it has been determined experimentally that sodium carbonate (inorganic electrolyte) readily reacted primarily with fine-grained clay particles contained in the clay soil Cl. This reaction increased the electrokinetic potential, which reduced the viscosity of the self-compacting grout. Chemically, this reaction resulted in the binding of polyvalent cations on the surface of the micelle of clay minerals. In this reaction, the sodium carbonate compound Na_2_CO_3_ was able to bind the Ca^2+^ cations contained in the clay soil and replace them with 2Na^+^ cations, which resulted in liquefaction of the self-compacting grout (by increasing the spillage value). For other grouts (Cl + 10% FBCA + 4% CEM + 0.1% SC = 690 mm, SF2, at 56% humidity; Cl + 30% FBCA + 4% CEM + 0.1% SC = 685 mm, SF2, at 62% humidity), FBCA and cement were more likely to affect spillage than sodium carbonate compared to the effect of sodium carbonate liquefying additive on spillage value. Based on the results of the spillage achieved, it can therefore be concluded that the use of the liquefying additive sodium carbonate is more suitable for clay soil Cl-based self-compacting grouts that do not contain FBCA.

It can be seen from Figure 6 that the compressive strength of the self-compacting clay soil Cl-based grout (Cl-REF) after seven days of maturation was not determined. This was because the samples were still too wet after seven days to determine the compressive strength value. Therefore, the results of compressive strength were compared primarily with self-compacting clay soil Cl-based grout and 4% cement and self-compacting clay soil Cl-based grout and 0.1% SC. When evaluating the compressive strength, it can be seen in Figure 6 that with the addition of 10% FBCA to clay soil Cl (Cl + 10% FBCA), the compressive strength value increased both after seven days (0.90 MPa) and 28 days of maturation (1.32 MPa) compared to grouts, where no fly ash was used (Cl + 4% CEM = 0.16 MPa after seven days of maturation and 0.38 MPa after 28 days of maturation). However, a higher dose of FBCA (30% of the weight of the soil) reduced the compressive strength values (0.79 MPa after seven days of maturation, 0.90 MPa after 28 days of maturation) of self-compacting grout (Cl + 30% FBCA). The decrease in compressive strength with increasing amount of fly ash was caused by the formation of less strong bonds between FBCA and clay soil Cl. It should be noted, however, that in the design of self-compacting grouts, the standard requirement of compressive strength was achieved after three days of maturation of more than 0.14 MPa and after 28 days of maturation of more than 0.2 MP, both when using 10% FBCA and also when using 30% fluid fly ash.

Based on a comparison of self-compacting grout of clay soil Cl and 4% cement (Cl + 4% CEM) with grouts containing 10% and 30% FBCA and additionally containing 4% cement, it can be stated that by increasing the cement dose, the compressive strength values still increased. With the addition of 10% FBCA and 4% cement to clay soil Cl (Cl + 10% FBCA + 4% CEM), a compressive strength of 0.95 MPa after seven days of maturation and 1.35 MPa after 28 days of maturation was achieved. On the other hand, a decrease in compressive strength was observed with the addition of 30% fluid fly ash and 4% cement to clay soil Cl (Cl + 30% FBCA + 4% CEM). Based on the results achieved, it can be stated that the development of compressive strength of self-compacting grouts was influenced both by the addition of FBCA (mainly in the amount up to 10% of the weight of the soil) and by the addition of 4% cement CEM II/BM (S-LL) 32.5 R.

An increase in compressive strength values was then observed for some self-compacting grouts with a subsequent addition of 0.1% of the sodium carbonate liquefying additive (of the weight of the soil). For self-compacting clay soil Cl-based grout with 10% FBCA, 4% cement, and 0.1% sodium carbonate, a compressive strength of 1.1 MPa was measured after seven days of maturation. However, after 28 days of maturation, this grout had a lower compressive strength (1.23 MPa) compared to the grout where sodium carbonate was not used (Cl + 10% FBCA + 4% CEM = 1.35 MPa, after 28 days of maturing). The development of compressive strength was influenced mainly by the moisture content in the grout already in the fresh state, which gradually went away and, thus, formed a more porous structure of self-compacting grout, and this had the effect of reducing the compressive strength. However, a more significant fact within the self-compacting grout structure was the finding that grouts containing FBCA (both 10% and 30% of the weight of the soil) in combination with sodium carbonate (0.1% of the weight of the soil) have a higher pore content. This was due to a chemical reaction between the FBCA and the sodium carbonate liquefied additive to form a more porous self-compacting grout structure. In particular, this significantly influenced the reduction of the compressive strength of the self-compacting grout containing 30% fluid fly ash in combination with 0.1% sodium carbonate. For this grout (Cl + 30% FBCA + 4% CEM + 0.1% SC), a compressive strength of 0.83 MPa after seven days of maturation and 0.93 MPa after 28 days of maturation was measured. Thus, these values are lower than in comparison with a self-compacting clay soil Cl-based grout with 10% FBCA, 4% cement, and 0.1% sodium carbonate.

Based on the achieved spillage and compressive strength values, it can be stated that for the improvement of these parameters, the use of FBCA in the amount of 10% (of the weight of the soil) in combination with sodium carbonate liquefying additive in the amount of 0.1% (of the weight of the soil) seems optimal for the design of self-compacting clay soil Cl-based grouts. When monitoring environmental parameters, it should be noted that even when using FBCA in the amount of 30%, these parameters were met.

### 3.2. Verification of Self-Compacting Grout in Situ

In the framework of research works, the designed optimal self-compacting grout consisting of clay soil Cl, 10% FBCA and 4% CEM II/B-M (S-LL) 32.5 R was verified in situ. In the case of this grout, the quality of the flow around installations, self-compaction, stability, and load capacity over time were monitored. The research work process was carried out at the construction site of an apartment building in Brno. The grout was tested on the excavation segment of 1.0 m × 0.6 m × 1.0 m, in which a polymer water pipe with a diameter of 125 mm was laid.

In the experiment clay soil Cl was used, which was excavated directly at the construction site. The cement CEM II/B-M (S-LL) 32.5 R and FBCA were brought to the construction site in bag form and stored in a dry environment. The cement and fly ash were dosed manually. All components were weighed using a precision scale before mixing.

Mixing of clay soil Cl with other components was carried out mechanically by means of a forced circulation mixer designed for soils (Figure 8a). The mixer is equipped with a sieve on its upper part for the separation of grains larger than 16 mm and possible organic impurities (roots, stalks of plants, etc.). The removal of larger grains was necessary because they could have made it difficult for the mixture to flow into the smallest spaces around the installation.

Initially, the clay soil Cl was mixed with cement and FBCA for five minutes. Subsequently, a small amount of water was added to the mixture. During intensive mixing, which took about five minutes, water was gradually added to the grout. The amount of water added to the grout was controlled by the optimum spillage value, which was verified by the Abrams cone slump-flow test with a measured spillage diameter of 680–700 mm (Figure 8b). After mixing, the grout was poured into the trench (Figure 8c). The grout was poured off directly into the trench (Figure 8d).

For laboratory testing, samples were taken from the grout for compressive strength determination after seven days storage under laboratory conditions (Figure 9a). It reached 0.7–0.95 MPa after seven days of maturation and 1.0–1.4 MPa after 28 days of maturation. In real conditions, it was observed that, over time, the grout became more bearable and after three days of maturing even fully walkable (Figure 9b and Figure 10a). At the same time, after three days of maturation, samples were taken at the construction site to verify further compressive strength tests (from Figure 10b to Figure 10d). The compressive strength of the mixture after three days of maturation was 0.3–0.5 MPa.

In terms of the assessment of the achieved properties, it can be stated that the grout easily flowed around the water installation and showed a self-compacting effect. The grout was, therefore, easy to pour off without the need for vibration. Due to the presence of cement and FBCA, the grout showed its stability. This phenomenon was observed by the fact that no sedimentation of the particles occurred, and the grout maintained a uniform dispersion of the particles until hardening. The effect of cement and FBCA was also manifested by increasing the load capacity of the grout over time.

## 4. Conclusions

This paper was focused on the research and development of the reuse of clay soil Cl, originating from the UU excavation works in the form of self-compacting grout. To achieve self-compacting grout parameters, clay soil Cl was mixed with FBCA, sodium carbonate, and water. The achieved results of research works are presented in the following points:The originality of the research of the article was to find an optimal technology or a new effective method that led to the transformation of clay soil into a self-compacting grout. Due to existing technologies (especially soil compaction), the research works had led to the development of the new and simple technology of self-compacting grouts. In comparison to existing utility excavation technology, it has been shown that these grouts can unequivocally compete with soil compaction due to their excellent properties that meet all standards, and without the need to transport to landfills.On the basis of the results obtained, it can be stated that by the appropriate choice of materials it is possible to achieve the treatment of clay soil Cl so that it can be reused in the trench and not transported to landfills.Chemical stabilization (achieved by a chemical reaction between the stabilizer/binder and minerals in the soil) and its subsequent liquefaction were used as an optimal treatment of clay soil Cl to achieve self-compacting grout parameters.For the chemical stabilization of clay soil Cl, FBCA was used in the amount of 10% and 30% (of the weight of the soil). The use of sodium carbonate in an amount of 0.1% (of the weight of the soil) was chosen as a suitable liquefying agent to achieve the self-compacting effect of the grout.It has been experimentally determined that the use of 10% FBCA with clay soil Cl is most suitable for achieving the optimal spillage parameter of self-compacting grout (class SF2 (660–750 mm) and class SF3 (760–850 mm)). For this self-compacting grout containing 10% fluidized fly ash, the highest spillage value of 710 mm (SF2) was achieved at 57% humidity. In assessing the effect of sodium carbonate on the spillage parameter, it has been found that its use is more suitable for self-compacting clay soil Cl-based grouts, which do not contain FBCA.In monitoring the effect of sodium carbonate on clay soil Cl, it was found that there was an increase in the electrokinetic potential, thereby reducing the viscosity of the self-compacting grout. Chemically, this reaction resulted in the binding of polyvalent cations on the surface of the micelle of clay minerals. The Na_2_CO_3_ sodium carbonate compound in this reaction was able to bind the Ca^2+^ cations contained in the clay soil and replace them with 2Na^+^ cations, which resulted in liquefaction of the self-compacting grout (by increasing the spillage value).From the point of view of the determination of the compressive strength parameter of self-compacting grouts, it was found, based on thorough laboratory testing, that the presence of FBCA and sodium carbonate in the grout, and also the amount of moisture of the self-compacting grout in the fresh state, have a significant effect on this parameter. To achieve optimum compressive strength value, the use of 10% FBCA and 4% cement with clay soil Cl seems to be suitable. The self-compacting grout thus designed has reached the highest compressive strengths (0.95 MPa after seven days of maturation and 1.35 MPa after 28 days of maturation).Based on the achieved spillage and compressive strength values, it can be stated that for the improvement of these parameters, when designing self-compacting clay soil Cl-based grouts, it is recommended to use FBCA in the amount of 10% (of the weight of the soil) in combination with a liquefying additive sodium carbonate in the amount of 0.1% (of the weight of the soil). Within the monitoring of environmental parameters, it should be noted that these parameters were also met when using FBCA in the amount of 30%.After verifying the effect of the self-compacting grout in situ, it can be stated that the grout easily flowed around the water installation and showed a self-compacting effect. The grout was therefore easy to pour off without the need for vibration. Due to the presence of cement and FBCA, the grout showed its stability. This phenomenon was observed by the fact that no sedimentation of the particles occurred, and the grout maintained a uniform dispersion of the particles until hardening. The effect of cement and FBCA was also manifested by increasing the load capacity of the grout over time.

## Figures and Tables

**Figure 1 materials-13-01972-f001:**
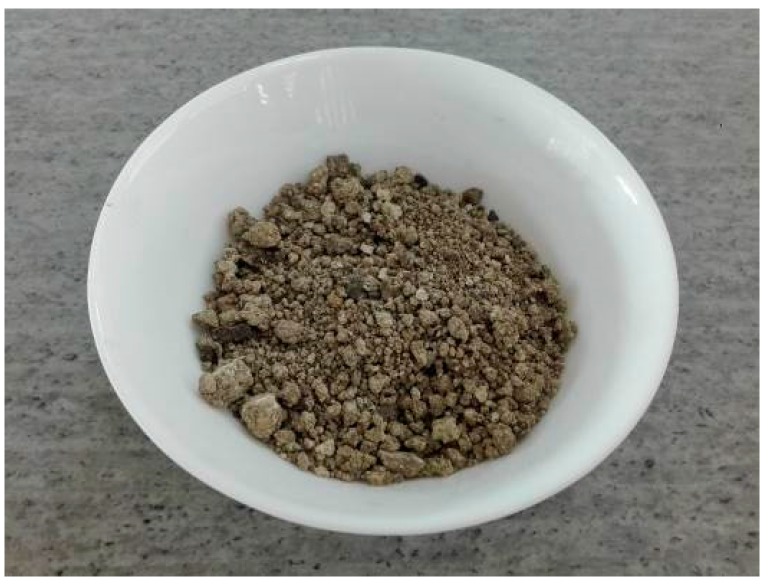
Clay soil ‘Cl’ from Central Europe (Brno, CZ).

**Figure 2 materials-13-01972-f002:**
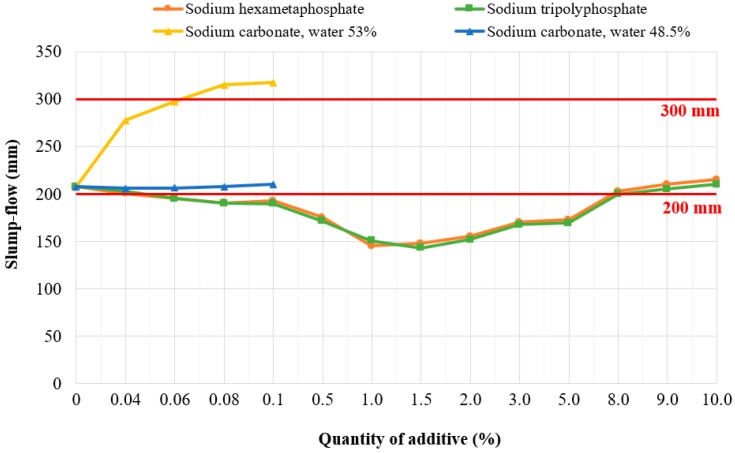
Verification of the effect of liquefying additives with clay soil ‘Cl’.

**Figure 3 materials-13-01972-f003:**
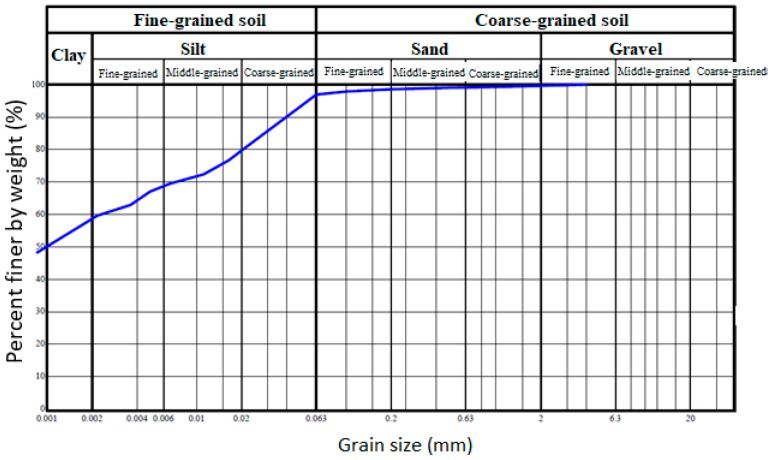
The granulation curve of soil from Central Europe (Brno, CZ), XRD analysis of clay soil ‘Cl’.

**Figure 4 materials-13-01972-f004:**
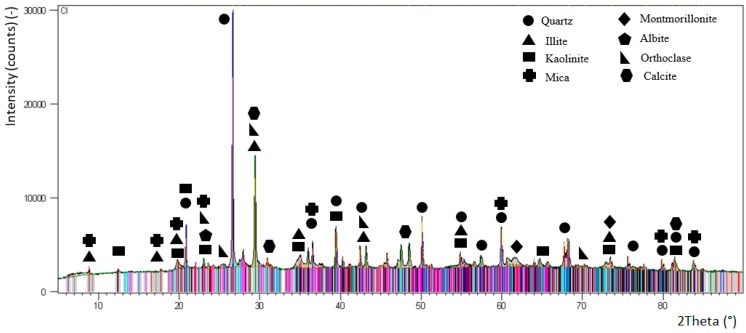
XRD analysis of clay soil ‘Cl’.

**Figure 5 materials-13-01972-f005:**
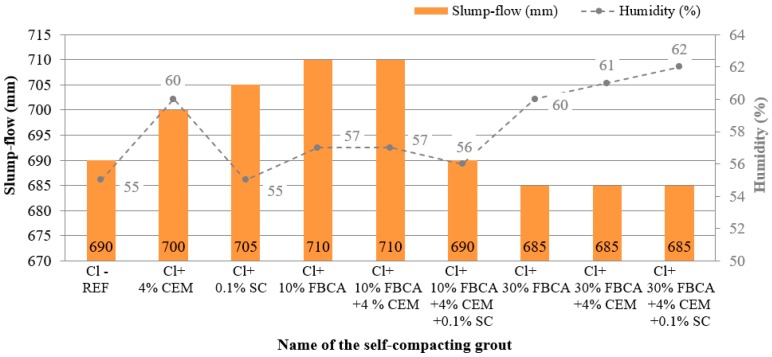
Slump-flow dependence on moisture of self-compacting grouts using FBCA and liquefying additive sodium carbonate.

**Figure 6 materials-13-01972-f006:**
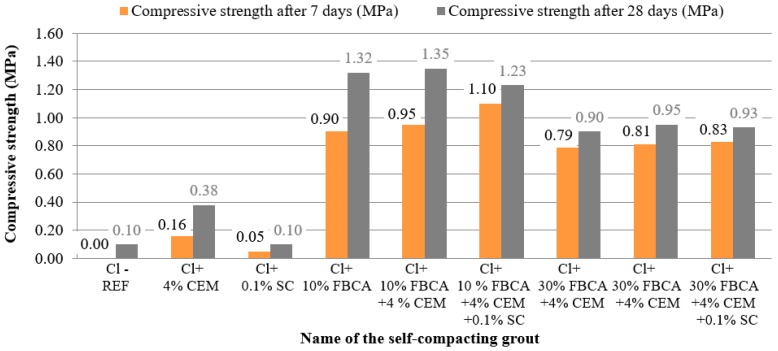
Compressive strengths of self-compacting grouts using FBCA and sodium carbonate (SC), after seven and 28 days of maturation.

**Figure 7 materials-13-01972-f007:**
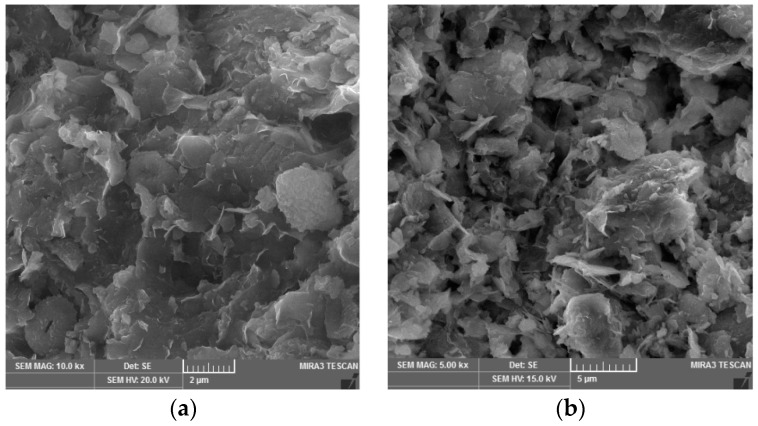
(**a**) SEM image of the self-compacting grout based on clay soil Cl, 4% CEM and 0.1% SC, (**b**) SEM image of the self-compacting grout based on clay soil Cl, 4% CEM, 10% FBCA, and 0.1% SC.

**Figure 8 materials-13-01972-f008:**
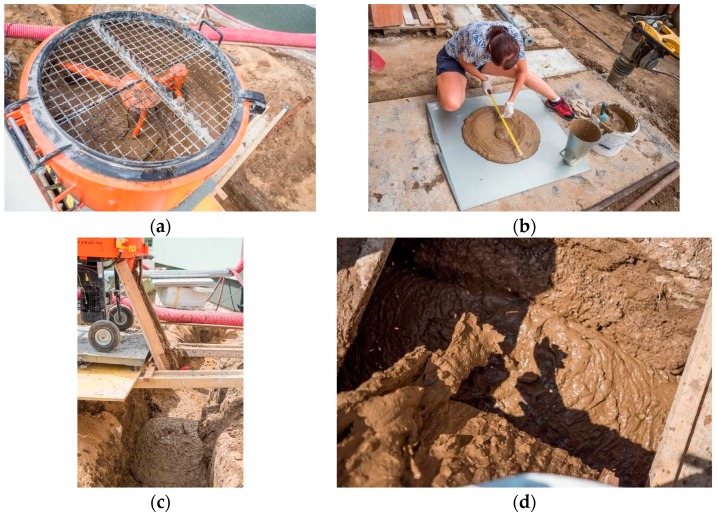
(**a**) Production of self-compacting grout; (**b**) spillage control to achieve a self-compacting effect; (**c**) filling the excavate by self-compacting grout; (**d**) filling the excavate by self-compacting grout.

**Figure 9 materials-13-01972-f009:**
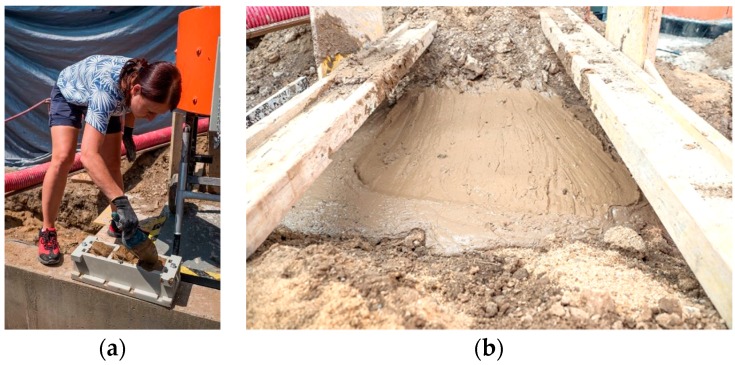
(**a**) Sampling of fresh grout for determination of the main parameters; (**b**) excavation filled with grout.

**Figure 10 materials-13-01972-f010:**
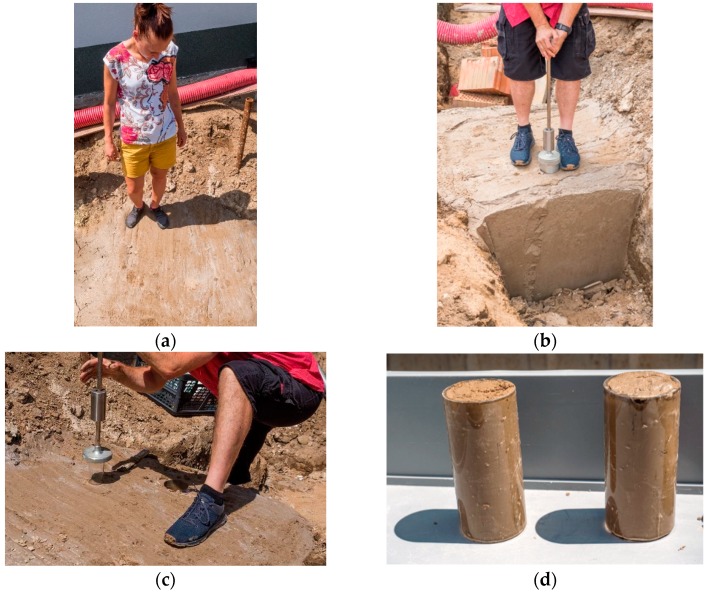
(**a**) Load capacity of the applied grout after three days; (**b**) sampling after three days from the implementation; (**c**) sampling after three days from the implementation; (**d**) samples taken for further testing.

**Table 1 materials-13-01972-t001:** Chemical composition of fluidized bed combustion fly ash (FBCA).

Input Material	SiO_2_ (%)	Al_2_O_3_ (%)	Fe_2_O_3_ (%)	SO_3_ (%)	CaO (%)	MgO (%)	K_2_O (%)	Na_2_O (%)	P_2_O_5_ (%)	Loss on Ignition (%)
FBCA	27.60	17.50	5.63	7.57	30.40	0.84	0.46	0.33	0.25	2.04

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
