# Peer review of "Utilization of Fluidized Bed Combustion Fly Ash in the Design of Reuse Clay Soil in the Form of Self-Compacting Grouts"

_materials, 2020, doi:10.3390/ma13081972_

Round 1

Reviewer 1 Report

This manuscript addresses a major civil engineering and construction issue in many European countries. This concerns the reuse of clay soils that arise in the framework of excavation works of underground utilities. This research was undertaken mainly because the production of unsuitable clay-rich soils from these utility excavations occurs very frequently and have to be disposed. In terms of all types of waste, it is about 65%. From a geotechnical point of view, it is impossible to obtain natural soil that meets the requirements for its re-use in the utilities excavation without any modification. In keeping with governmental requirements to improve the quality of construction the goal was to find a solution to the problem, not only by minimizing the formation of these soils, but also in their use for subsequent building production. The authors used soils that occur on Jurassic and Cretaceous sediments in central Europe. For the mechanical treatment of this clay soil they decided to use an admixture of other soils or materials such as fly ash of suitable grain size and moisture. The purpose was so an improvement in moisture and an adjustment of grain size distribution would be achieved. As they point out, the percentage of the individual components in the soil has a fundamental influence on its mechanical properties. For this reason, a substance known as fluidized bed combustion fly ash was used as an additional raw material. With regard to monitoring the influence of fly ash on soil properties adjustment, resulting self-compacting grout and environmental aspect, the use of fly ash in an amount of 10% and 30% was chosen, based on soil weight. The authors conducted an extensive range of mineralogical and geotechnical analyses on the prepared samples, and found that the development of compressive strength of self-compacting grouts was influenced both by the addition of the fly ash and by the addition of 4% cement. Further, the authors found that the effect of sodium carbonate added to the clay soil was an increase in the flow of electricity potential, thereby reducing the viscosity of the self-compacting grout. Chemically, this reaction resulted in the binding of polyvalent cations on the surface of the clay mineral particles. This is an excellent piece of research that is certain to provide helpful information throughout the industry dealing with underground utilities. The authors have done a very thorough job and have explained clearly and with excellent illustrations the results of their work. The references are excellent and up to date. So I am pleased to recommend that this manuscript is acceptable for publication. I have two, very minor editorial suggestions to make to the authors:

1. Line 28 might read, “This paper researches the usability of clay soils……….”

2. Line 154 might read, “According to the EN 12350-8 the class values of slump-flow (SF) test are divided…….”

Author Response

This manuscript addresses a major civil engineering and construction issue in many European countries. This concerns the reuse of clay soils that arise in the framework of excavation works of underground utilities. This research was undertaken mainly because the production of unsuitable clay-rich soils from these utility excavations occurs very frequently and have to be disposed. In terms of all types of waste, it is about 65%. From a geotechnical point of view, it is impossible to obtain natural soil that meets the requirements for its re-use in the utilities excavation without any modification. In keeping with governmental requirements to improve the quality of construction the goal was to find a solution to the problem, not only by minimizing the formation of these soils, but also in their use for subsequent building production. The authors used soils that occur on Jurassic and Cretaceous sediments in central Europe. For the mechanical treatment of this clay soil they decided to use an admixture of other soils or materials such as fly ash of suitable grain size and moisture. The purpose was so an improvement in moisture and an adjustment of grain size distribution would be achieved. As they point out, the percentage of the individual components in the soil has a fundamental influence on its mechanical properties. For this reason, a substance known as fluidized bed combustion fly ash was used as an additional raw material. With regard to monitoring the influence of fly ash on soil properties adjustment, resulting self-compacting grout and environmental aspect, the use of fly ash in an amount of 10% and 30% was chosen, based on soil weight. The authors conducted an extensive range of mineralogical and geotechnical analyses on the prepared samples, and found that the development of compressive strength of self-compacting grouts was influenced both by the addition of the fly ash and by the addition of 4% cement. Further, the authors found that the effect of sodium carbonate added to the clay soil was an increase in the flow of electricity potential, thereby reducing the viscosity of the self-compacting grout. Chemically, this reaction resulted in the binding of polyvalent cations on the surface of the clay mineral particles. This is an excellent piece of research that is certain to provide helpful information throughout the industry dealing with underground utilities. The authors have done a very thorough job and have explained clearly and with excellent illustrations the results of their work. The references are excellent and up to date. So I am pleased to recommend that this manuscript is acceptable for publication. I have two, very minor editorial suggestions to make to the authors:

1. Line 28 might read, “This paper researches the usability of clay soils……….”

2. Line 154 might read, “According to the EN 12350-8 the class values of slump-flow (SF) test are divided…….”

We thank the reviewer for a concise summary of our manuscript, which we highly appreciate. We also thank for the valuable recommendations, which we respected and corrected. Base on the recommendations, the following sentence corrections were made:

1. Line 28 might read, “This paper researches the usability of clay soils……….”

2. Line 154 might read, “According to the EN 12350-8 the class values of slump-flow (SF) test are divided…….”

Reviewer 2 Report

In manuscript the authors presented influence of fluidized bed combustion fly ash (FBCA) and liquefying additives on the formation of structure and on the resulting properties of self-compacting grouts based on clay soil. In order to establish the influence of individual materials, researches were realized on samples without the fly ash. The mixture was obtained from clay soil + cement and fluidized bed combustion fly ash (10% to 30%) and liquefying additives. Properties of the designed self-compacting grout were verified by spillage testing and compressive strength. The authors demonstrated that fluidized bed combustion fly ash and liquefying additive have a significant influence on the formation of the structure of the self-compacting grout, and due to their presence, the compressive strength of the samples increased up to 0.5 MPa after seven days of hardening.

Observation:

The title is clear.

The subject matter is within the scope of the journal

This article contains new aspects; the authors must underline the major findings of their work and explain how the use of their proposed materials represents a progress to other similar published papers. Please point clears the originality.

The manuscript adheres to the journal's standards after revision

The Abstract section should refer to the study findings, methodologies, discussion as well as conclusion. Please rewrite Abstract, in this form is too generally.

The key words permit found article in the current registers or indexes. Please used only significantly keywords.

In the introduction it is relatively clearly described the state of the art of the investigated problem. More references form 2020 are necessary.

Please revise the chapter: 2. Materials and Methods, the experimental results must be presented in Results and discussion.

In abstract authors presented: ….”liquefying additive (sodium carbonate 0.1%) were used as …”, but in Figure 2 were presented more additives: Figure 2. Verification of the effect of liquefying additives with clay soil ‘Cl’, please clarify this non concordance.

The figures have good quality. I don’t know if is possible to use people photo without permission (in my country is forbidden, personal data protection).

Please put the figures in the order.

In table have necessary results.

The manuscript is relatively easy to understand by scientists form other area.

Comparisons with other researches must be presented.

The conclusion is OK. The conclusions were been sufficiently justified.

Please present references from 2020.

Please put axes in Fig 1(b).

Please provide minimum 2 references from this journal (last years), for demonstrated that manuscript is Materials topic.

Please verify all references

The paper was written in standard, grammatically correct English, but more corrections are necessary.

For example:

This paper researches of the usability…. Not paper researches, the author researched …

In terms of all types of waste, it is about 65% [1-3] …. This sentence is not clear!

… In view of the coming new legislative requirements, and especially under the Waste Framework Directive [4], …. These new legislative requirement are from 2008, so not new.

etc.

Please verify carefully all manuscript!

Please respect Guide for authors! It is necessary to verify references.

Author Response

Thanks reviewer for the valuable comments, which we are aware of. We respected all the comments and corrected them. Based on the reviewer's comments, the following corrections have been made:

1. This article contains new aspects; the authors must underline the major findings of their work and explain how the use of their proposed materials represents a progress to other similar published papers. Please point clears the originality.

It is not clear from the reviewer where the explanation of originality should be added, that's why the following text was added to the conclusion:

"The originality of the research of the article was to find an optimal technology or a new effective method that led to the transformation of clay soil into a self-compacting grout. Due to existing technologies (especially soil compaction), the research works had led to the development of the new and simple technology of self-compacting grouts. In comparison to existing utility excavation technology, it has been shown that these grouts can unequivocally compete with soil compaction due to their excellent properties that meet all standards. And without the need to transport to landfills."

2. The Abstract section should refer to the study findings, methodologies, discussion as well as conclusion. Please rewrite Abstract, in this form is too generally.

The abstract was rewritten to include the study findings, methodologies, discussion as well as conclusion.

3. The key words permit found article in the current registers or indexes. Please used only significantly keywords.

Keywords were corrected based on the reviewer's comments.

4. In the introduction it is relatively clearly described the state of the art of the investigated problem. More references form 2020 are necessary.

References have been updated to current validity. At the same time, additional resources were added.

5. Please revise the chapter: 2. Materials and Methods, the experimental results must be presented in Results and discussion.

The chapter 2. was revided and the experimental results were removed to the chapter 3. Results and discussion.

6. In abstract authors presented: ….”liquefying additive (sodium carbonate 0.1%) were used as …”, but in Figure 2 were presented more additives: Figure 2. Verification of the effect of liquefying additives with clay soil ‘Cl’, please clarify this non concordance.

An explanation of the use of all other additives was added to the text.

7. The figures have good quality. I don’t know if is possible to use people photo without permission (in my country is forbidden, personal data protection).

On the figures are view only the authors of the article who agree to their publication.

8. Comparisons with other researches must be presented.

The originality of the article has been added to the text. All results in this article are based only on our own research, which we consider as unique. Comparison with other research would lead to the creation of another article. There is currently nothing to compare the article with.

9. Please present references from 2020.

It has been done.

10. Please put axes in Fig 1(b).

It has been done.

11. Please provide minimum 2 references from this journal (last years), for demonstrated that manuscript is Materials topic.

It has been done.

12. Please verify all references.

It has been done.

13. The paper was written in standard, grammatically correct English, but more corrections are necessary.

The article was reviewed by a native speaker. If you want, we will send you a certificate of the revision.

14. … In view of the coming new legislative requirements, and especially under the Waste Framework Directive [4], …. These new legislative requirement are from 2008, so not new.

This legislative document has been changed to a valid one.

15. Please respect Guide for authors! It is necessary to verify references.

References were verified according to Guide for authors and it was found that they should be correct. In references were done only small corrections.

Round 2

Reviewer 2 Report

Utilization of fluidized bed combustion fly ash in the design of reuse clay soil in the form of self-compacting grouts

authors Rostislav Drochytka, Magdaléna Michalčíková*

In manuscript the authors presented influence of fluidized bed combustion fly ash (FBCA) and liquefying additives on the formation of structure and on the resulting properties of self-compacting grouts based on clay soil. In order to establish the influence of individual materials, researches were realized on samples without the fly ash. The mixture was obtained from clay soil + cement and fluidized bed combustion fly ash (10% to 30%) and liquefying additives. Properties of the designed self-compacting grout were verified by spillage testing and compressive strength. The authors demonstrated that fluidized bed combustion fly ash and liquefying additive have a significant influence on the formation of the structure of the self-compacting grout, and due to their presence, the compressive strength of the samples increased up to 0.5 MPa after seven days of hardening.

The title is clear.

The subject matter is within the scope of the journal

This article contains new aspects; the authors underline the major findings of their work and explain how the use of their proposed materials represents a progress to other similar published papers.

The manuscript adheres to the journal's standards.

The Abstract was rewrite.

The key words permit found article in the current registers or indexes. The keywords were verified.

In the introduction is clearly described the state of the art of the investigated problem.

The figures have good quality.

In table have necessary results.

The manuscript is easy to understand by scientists form other area.

The conclusion is OK. The conclusions were been sufficiently justified.

The paper was written in standard, grammatically correct English, more corrections were made.

Accept in this form